# ctDNA and tumor-based biomarkers of giredestrant response in acelERA breast cancer

Ann E. Collier [1]✉, Stephanie Hilz[1], Alejandro M. Chibly [1], Chunzhe Duan[2], Lincoln W. Pasquina [3], Xiaopeng Sun[1], Mariana Chavez-MacGregor [4], Aditya Bardia [5], Miguel Martín [6], Elgene Lim [7], Joohyuk Sohn [8], Pablo Diego Pérez-Moreno[1], Tharu M. Fernando [1] & Heather M. Moore[1]

Endocrine therapy (ET) resistance in estrogen receptor positive (ER+) advanced breast cancer is often linked to *ESR1* mutations, yet responses to oral selective ER degraders vary within mutant subgroups. Through a biomarker analysis of acelERA Breast Cancer (NCT04576455), we show that tumor ER transcriptional activity as well as circulating tumor DNA (ctDNA) genomics and dynamics effectively stratify response to ET, including giredestrant. We find that following first-line therapy, the ctDNA genomic landscape is diverse and influenced by CDK4/6 inhibitor exposure. Despite this complexity, ER activity in *ESR1*-mutant tumors remains comparable to early breast cancer but is reduced in most non-mutant cases. This maintained ER activity is associated with giredestrant benefit. Furthermore, early ctDNA clearance identifies responding patients, and the combination of low ER activity and high ctDNA burden predicts rapid clinical progression. These findings provide a framework for personalizing future breast cancer therapies by integrating liquid biopsies with tissue-based signatures.

Approximately 70% of diagnosed breast cancer cases are estrogen receptor-positive (ER+) and are largely fueled by estrogen[1,2]. Therefore, targeting the ER pathway has been a successful strategy for therapeutic intervention for a large proportion of patients with breast cancer[3]. Despite notable advances, resistance to endocrine therapy (ET) often emerges, necessitating the development of new therapeutics such as selective estrogen receptor antagonists and degraders (SERDs)[4–6]. Oral SERDs have recently emerged as promising substitutes to currently approved ETs[7–9] and the intramuscularly injected SERD fulvestrant, offering potential advantages in efficacy, with improved administration methods and bioavailability. Despite these possible advantages, data on oral SERD efficacy and associated biomarkers remains limited.

Mutations in the estrogen receptor 1 (*ESR1*) gene, which encodes the ER alpha protein, have emerged as potential biomarkers of ET resistance as well as responsiveness to oral SERDs[10]. These alterations, enriched in the advanced setting and most commonly acquired following aromatase inhibitor (AI) therapy, confer ligand-independent receptor activation to promote breast cancer progression[10,11]. Recent phase 2 and 3 clinical trials in advanced breast cancer (aBC) have consistently shown that patients with *ESR1* mutant (*ESR1*m) tumors generally show a more pronounced response to single agent oral SERDs compared to standard-of-care ETs[12–15].

Here, we present an exploratory biomarker analysis of the phase 2 acelERA Breast Cancer (BC) study (NCT04576455), which compared

[1]Genentech, Inc., South San Francisco, CA, USA. [2]Roche (China) Holding Ltd., Shanghai, China. [3]Foundation Medicine Inc., Boston, MA, USA. [4]The University of Texas MD Anderson Cancer Center, Houston, TX, USA. [5]University of California, Los Angeles, CA, USA. [6]Hospital Gregorio Marañón, Universidad Complutense, GEICAM, CIBERONC, Madrid, Spain. [7]Garvan Institute of Medical Research, St Vincent's Clinical School, University of New South Wales, Darlinghurst, Australia. [8]Division of Medical Oncology, Department of Internal Medicine, Yonsei University College of Medicine, Seoul, South Korea. ✉e-mail: collier.annie@gene.com

giredestrant to physician's choice of ET (PCET) in 2L/3L ER+, HER2− locally advanced or metastatic breast cancer (La/mBC) patients. Giredestrant is a highly potent, nonsteroidal, orally bioavailable small molecule SERD and full ER antagonist that achieves robust ER occupancy and is designed to drive deep and sustained inhibition of ER signaling, both ligand-dependent and ligand-independent[16–18]. Giredestrant competitively binds wild-type and mutant ER, blocking coactivator recruitment and immobilizing the receptor to suppress activity and reduce protein levels. As previously reported[15,19], acelERA BC did not reach statistical significance for the primary endpoint of investigator-assessed progression-free survival (INV-PFS), but giredestrant exhibited a numerical improvement over PCET as noted by ~20% relative reduction in risk of progression or death and favorable results for secondary efficacy endpoints. Giredestrant also demonstrated a more pronounced benefit in patients harboring detectable *ESR1*m, with a hazard ratio (HR) of 0.60 (95% CI, 0.35–1.03). This study aims to uncover additional indicators of ET (giredestrant, fulvestrant, and AI) response, including biomarkers derived from circulating tumor DNA (ctDNA). ctDNA has been demonstrated as a valuable noninvasive predictive and prognostic biomarker across many indications[20], providing insights into tumor heterogeneity[14,21,22], therapy response monitoring[23–25], relapse detection[26,27], and identification of resistant cell populations[26]. Exploratory biomarker analyses from this study will aid in the development of future strategies for patient selection, clinical trial design, and treatment monitoring for oral SERDs.

## Results

### Baseline ctDNA landscape of 2L/3L ER+ HER2− La/mBC patients

Baseline plasma from 229 of 303 acelERA patients was evaluated with FoundationOne® Liquid CDx (F1LCDx), which sequences >1.25 Mb of genomic content covering 324 genes (Fig. 1a). Additional F1LCDx testing was performed at cycle 2 day 1 (C2D1, $n = 220$) and end-of-treatment (EOT, $n = 155$). Most PCET arm patients (~75%) received fulvestrant, with the remainder receiving an AI. For subjects enrolled in China, the PredicineCARE assay was used to determine *ESR1*m status ($N = 3$). The following analyses focus exclusively on F1LCDx results.

ctDNA tumor fraction (TF) was defined as the proportion of cell-free DNA originating from cancer cells relative to the total amount of cell-free DNA present in a given plasma sample. At baseline, ctDNA was detected in 78% (179) of evaluable samples, with a median TF of 7.7% overall and 14% in patients with samples above the limit of detection (Fig. 1a, b, not determined with PredicineCARE assay). There was no significant difference in baseline ctDNA TF levels or *ESR1*m allele frequency (MAF) between treatment arms, regardless of *ESR1*m status (Fig. 1b, Supplementary Fig. 2a). Among the samples where ctDNA was detected, 98% (176) harbored at least one pathogenic (known or likely to affect protein function) SV, the majority of which were substitutions of unknown significance (Fig. 1c, d). Pathogenic mutations in *ESR1* were the most prevalent alterations detected (49.2%), followed by *PIK3CA* (48.6%), *TP53* (37.4%), and *RB1* (15.1%) (Fig. 1e). Copy number alterations were detected in 36% of samples, primarily involving *FGF19/3/4*, *CCND1*, and *FGFR1*, likely from the same amplicon (Supplementary Fig. 1a, b). Pathogenic rearrangements were observed in 16% of samples, most commonly as one or two large truncations or deletions not enriched for a particular gene. (Supplementary Fig. 1c, d). *ESR1* and *PIK3CA* SVs co-occurred in ~15% of samples with F1LCDx results (Supplementary Fig. 1e).

We identified 183 pathogenic *ESR1* mutations (known or likely to affect ER protein function) in 88 patients. The most common variants were Y537X (X denotes any amino acid, 64%) and D538G (60%), E380Q (22%), and L536X (22%) with 54% of patients harboring multiple *ESR1* mutations (Fig. 1f, g, Supplementary Fig. 2c). Most *ESR1* mutations (73%) were subclonal (Fig. 1h). There was no difference in the number of *ESR1*m per patient based on whether a patient had received 1 or 2

prior lines of therapy (Supplementary Fig. 2d). Patients with *ESR1*m also had a lower prevalence of *PIK3CA* and *CDH1* pathogenic alterations (Fig. 1i).

In the study cohort, 41% (125/303) of patients received a prior CDK4/6 inhibitor (i), while the remainder had received prior AI or other ET without CDK4/6i exposure. As 1st line treatment with CDK4/6 inhibitors in the advanced setting has been shown to result in altered genomic profiles compared to AI alone[11], this presented a unique opportunity to isolate specific biomarkers associated with CDK4/6i resistance. Prior CDK4/6i treatment was associated with *ESR1* mutations, which were twice as prevalent in CDK4/6i-experienced patients and also greater in number per patient compared to CDK4/6i-naive patients (Fig. 1j, k). *RB1* alterations were three times more common post-CDK4/6i, consistent with known resistance mechanisms[28] (Fig. 1j). *ESR1* mutant allele frequency (MAF) appeared consistent regardless of prior CDK4/6i treatment (Fig. 1l), but ctDNA rates were higher in patients with prior CDK4/6i therapy (Fig. 1m, n).

### ctDNA biomarkers associated with response to endocrine therapy

Previous work examined the association of INV-PFS with baseline demographics in acelERA[15]. Consistent with other studies evaluating oral SERDs[12–14], we found that the presence of an *ESR1*m correlated with an improved response to giredestrant versus PCET (Fig. 2a, Supplementary Fig. 3a). This benefit was statistically significant for patients with disease harboring a single *ESR1*m, and showed a similar trend with multiple *ESR1* mutations (independent of ctDNA tumor fraction, Fig. 2b, Supplementary Fig. 3a). Giredestrant showed PFS improvement over PCET if a D538G (HR 0.49, mPFS 5.4 vs. 2.8 months) or Y537X (HR 0.60, mPFS 4.3 vs. 2.8 months) *ESR1* variant was detected (Supplementary Fig. 3a, note that many patients had more than one variant).

When looking specifically in patients with *ESR1*m disease, giredestrant trended toward a favorable benefit in many subgroups, particularly patients with *PIK3CA*m tumors, high tumor ER transcriptional activity, or patients with no prior CDK4/6i exposure (Supplementary Fig. 3b). As such, the acelERA trial protocol included both *ESR1*m status and prior CDK4/6i treatment as stratification factors, recognizing their potential impact on treatment outcomes. As reported previously, giredestrant did not show significant improvement over PCET based on a patient's prior CDK4/6i exposure and acelERA was not powered to ask this question[15]. However, when looking across subgroups based on both *ESR1*m and CDK4/6i experience, giredestrant showed a near 3-fold PFS improvement over PCET among patients with *ESR1*m, CDK4/6i naive tumors (HR 0.34, mPFS 11.1 vs. 3.8 months, Fig. 2c, d, Supplementary Fig. 3a, b), which likely have less evolved resistance mechanisms and reduced genomic complexity/clonal heterogeneity compared to CDK4/6i-exposed patients. Within the giredestrant arm, outcomes were similar regardless of *ESR1*m status (Fig. 2e).

Given that *ESR1* mutations alone did not fully predict treatment response, we investigated whether additional genomic features and ctDNA levels might better identify patients likely to benefit from ET. Beyond *ESR1*m, we found baseline ctDNA detection and high TF levels were prognostic regardless of treatment arm (Supplementary Fig. 4a–c). Baseline ctDNA TF was significantly higher in patients who had confirmed progressive disease (PD) (Fig. 2f). All partial responses (PRs) in the PCET arm occurred only when ctDNA was undetectable, whereas giredestrant produced PRs across a range of tumor fractions (0–35%, median 7%, Fig. 2f). In either arm, PFS was improved nearly 4-fold in patients without detectable ctDNA (HR 0.33, mPFS 12.75 vs. 3.81 months) (Fig. 2g, Supplementary Fig. 4a–c) and patients with high baseline TF (>20%) had particularly poor outcomes (Fig. 2h, Supplementary Fig. 4a–c).

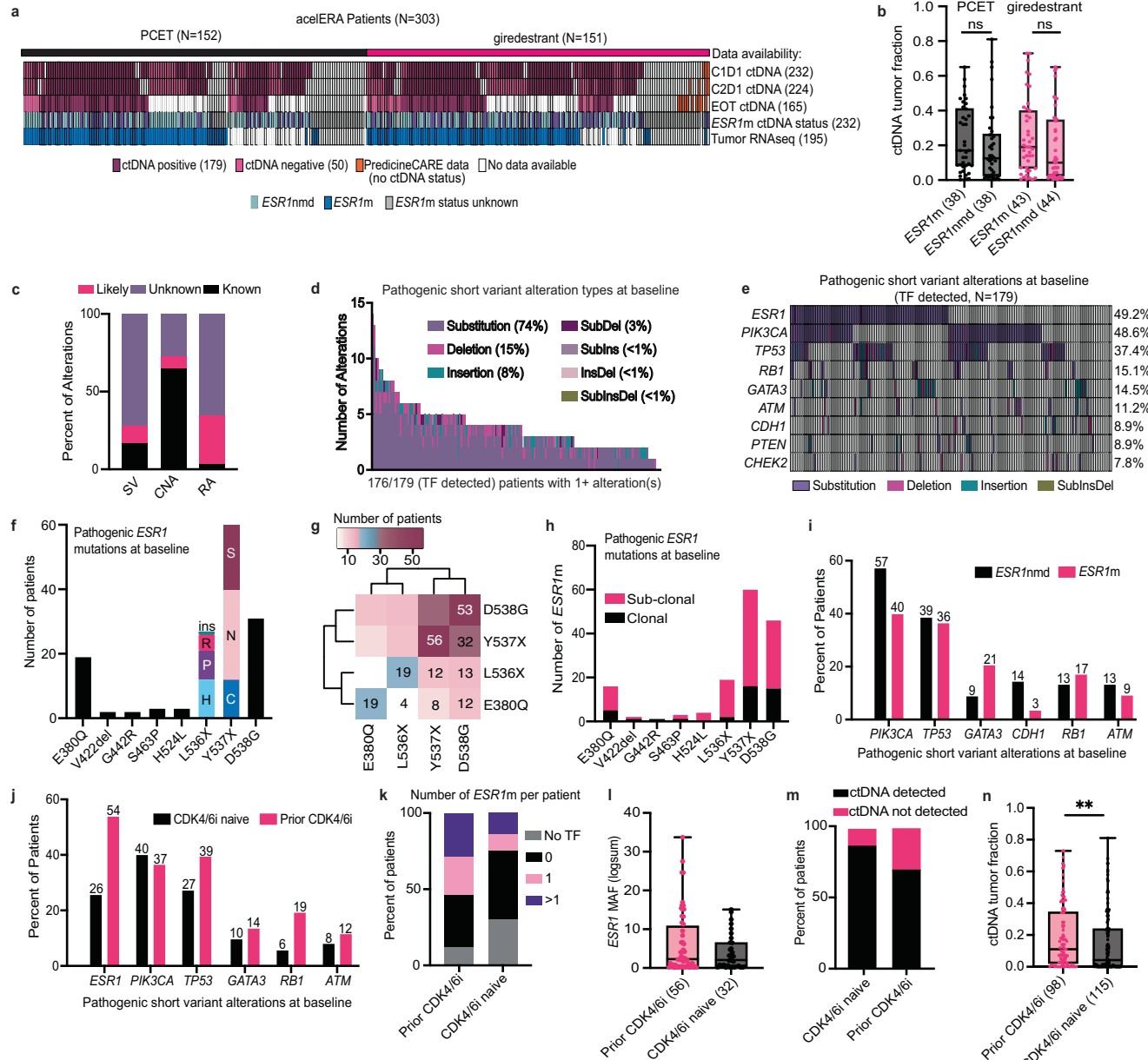

**Fig. 1 | Baseline genomic landscape and ctDNA characteristics of acelERA 2/3L ER+ HER2- La/mBC patients. a** Summary of evaluable biomarker data (*N* = 303 patients); columns represent individual patients. *ESR1*m status was defined by baseline plasma ctDNA alterations known or likely to affect ER protein function. **b** Baseline ctDNA tumor fraction (TF) by treatment arm and *ESR1*m status (*p* = 0.2053, 0.1502). **c** Distribution of pathogenic significance for all alterations detected by F1LCDx. **d** Summary of pathogenic (known or likely) short variants identified by F1LCDx at baseline, colored by alteration class. **e** Prevalence of the most common pathogenic short variants; columns represent patients with detectable ctDNA TF. **f** Prevalence of each *ESR1* pathogenic variant detected in the acelERA study cohort. **g** Co-occurrence matrix for the four most common *ESR1* variants. The number of samples with each combination of variants is indicated in the respective box. **h** Distribution of *ESR1* variants by clonal status (sub-clonal

defined as MAF/TF < 0.25) **i** Prevalence of common pathogenic short variants stratified by *ESR1*m status or **j** prior CDK4/6i exposure, in patients with detectable ctDNA TF. **k** Number of *ESR1*m detected per patient, **l** baseline *ESR1* MAF in each arm, **m** baseline ctDNA detection, or **n** baseline TF, split by prior CDK4/6 exposure, *p* = 0.0095. PCET physician's choice endocrine therapy, m mutation, nmd no mutation detected, TF tumor fraction, MAF mutant allele frequency, F1LCDx FoundationOne Liquid CDx, CDK4/6i cyclin-dependent kinase 4/6 inhibitor, SV short variant, CNA copy number alteration, RA rearrangement alteration. Box plots represent median and interquartile range (IQR), with whiskers depicting minimum and maximum. Two-sided Mann–Whitney tests were used for group comparisons (*p*-values indicated); no multiple-comparison adjustment was performed. Source data are provided as a Source Data file.

Additional genomic features showed treatment-specific associations. *RB1* mutations predicted poor response regardless of treatment, while *ESR1* mutations were negatively associated with PCET response (Supplementary Fig. 4a, b). Giredestrant outperformed PCET in patients with *PIK3CA*m disease or tumors without detectable *RB1* mutations (Supplementary Fig. 3a). Patients with co-occurring *ESR1*m and *PIK3CA*m disease (*n* = 35) saw particularly improved PFS with

giredestrant (HR 0.38, mPFS 5.6 vs. 1.8 months). These results highlight the prognostic value of baseline ctDNA levels and identify specific genomic features predictive of response to different ETs.

Given the robust prognostic value of ctDNA, we examined clinical features associated with the 22% (50) of patients without detectable baseline ctDNA. These patients were more likely to be non-white, have higher BMI, no liver metastases, lower LDH, and no prior CDK4/6i

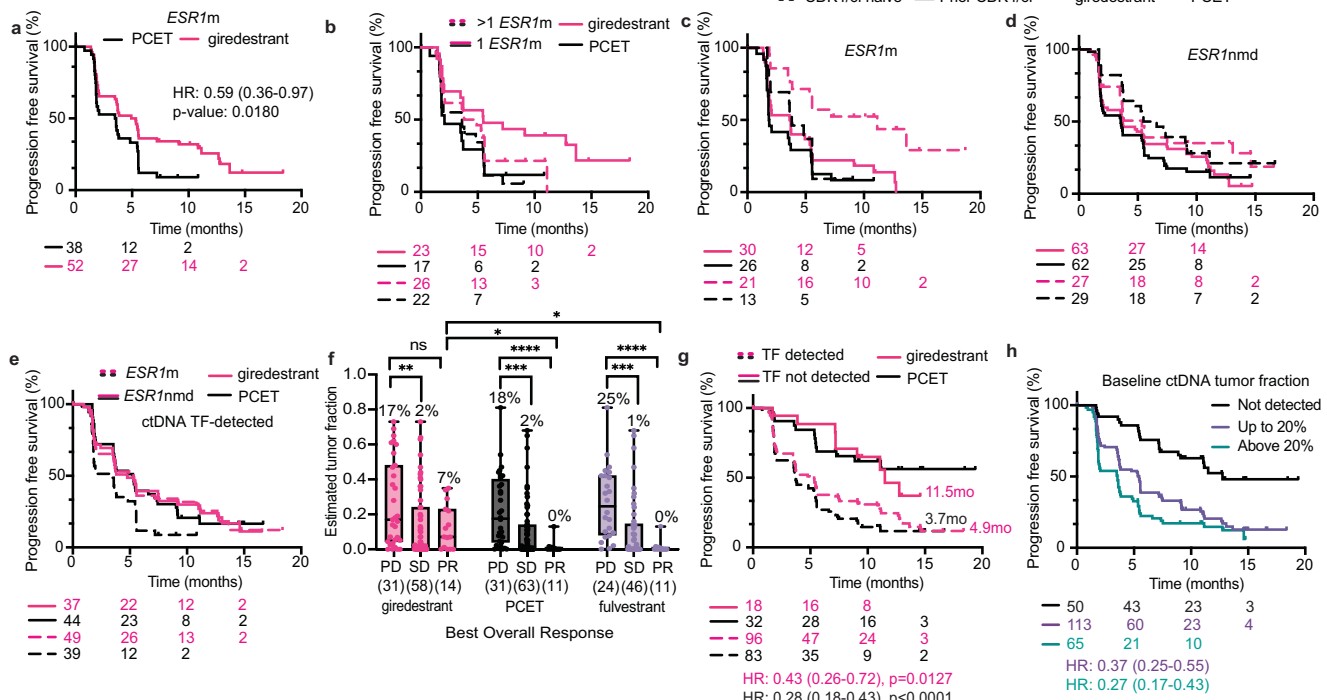

**Fig. 2 | ctDNA biomarkers associated with endocrine therapy response and PFS outcomes. a** Progression free survival (PFS) stratified by *ESR1*m status or **b** the number of detectable *ESR1* mutations. **c** PFS according to prior CDK4/6i exposure in patients with detectable *ESR1*m or **d** *ESR1*nmd baseline ctDNA. Includes only patients with detectable ctDNA tumor fraction (TF). **e** PFS for patients with detectable TF according to *ESR1*m status. **f** Baseline ctDNA TF split by treatment arm and confirmed best overall response, *$p = 0.0135$, **$p = 0.0045$, ***$p = 0.0002$, ****$p < 0.0001$. **g** PFS according to ctDNA TF detection at baseline ($p = 0.0127$ for giredestrant and $p < 0.0001$ for PCET) or **h** the level of ctDNA TF detected at

baseline. PCET physician's choice endocrine therapy, HR hazard ratio, CI confidence interval, m mutation, nmd no mutation detected, TF tumor fraction, PD progressive disease, SD stable disease, PR partial response, mo months. Box plots represent median and interquartile range (IQR), with whiskers depicting minimum and maximum. Two-sided Mann–Whitney tests were used for group comparisons ($p$-values indicated). PFS $p$-values were determined by two-sided log-rank (Mantel–Cox) tests. Hazard ratios (HR) and 95% confidence intervals (CI) were calculated using the log-rank approach. Source data are provided as a Source Data file.

exposure (Supplementary Fig. 5a). Only 26% of ctDNA-negative patients had prior CDK4/6i exposure, compared to 51% of ctDNA-positive patients. Tumors from patients with detectable ctDNA ($N = 134$), with presumably higher rates of ctDNA shedding, had lower levels of estrogen response signatures compared to those without detectable TF ($N = 35$, Supplementary Fig. 5b). We found that ctDNA detection rates, PFS outcomes, and on-treatment TF dynamics were similar between ductal and lobular breast cancer (Supplementary Fig. 6a–e), supporting ctDNA's utility across histological subtypes.

## Tumor ER activity is a predictor of giredestrant response

Given that gene expression patterns can reveal functional pathway activities not captured by mutation status alone, we next examined tumor transcriptional profiles. RNAseq was performed on fresh or archival tumor tissue from 195 patients, collected prior to the initiation of study therapy (Fig. 1a). Of note, the majority of patients enrolled in the study were considered to have "ER positive" (>10% cells) tumors, and only 7/303 were "ER low" (1–10% cells) based on prior immuno-histochemistry (IHC). First, without considering prior therapies of each tumor, we found that the early estrogen response hallmark pathway was significantly correlated with giredestrant PFS (Supplementary Fig. 7a). ER transcriptional activity, quantified by measuring the expression of known E2 repressed and induced genes[18], was significantly higher in tumors that exhibited stable disease (SD) or PR with giredestrant (Fig. 3a); PCET showed a similar direction of effect but did not reach significance. A non-significant ER activity × treatment arm interaction ($p = 0.946$ in Cox model and $p = 0.344$ in two-way ANOVA) suggests ER activity's predictive effect may apply across ETs, though this study was not powered for interaction analysis. Patients with high

ER tumor activity (z-score >0) or more luminal-like tumors (defined by TCGA-derived transcription factor scores[29]) also showed longer PFS with giredestrant (median 7.5 vs 3.6 months; Fig. 3b). The PCET arm showed a similar but non-significant separation (5.6 vs 5.1 months; Supplementary Fig. 7b). Tumors with high ER activity also had increased expression of cell cycle genes (Supplementary Fig. 7c). Thus, elevated ER activity functions as a prognostic marker of endocrine sensitivity and may predict patients most likely to benefit from giredestrant.

A key caveat is that many tumors were scored from archival biopsies; if ER activity declined during intervening treatments, some "ER-high" cases were probably mis-classified as such, dampening the true benefit signal and confounding any direct comparison between giredestrant and PCET. We therefore categorized tissue samples based on biopsy timing: 97 tumors were treatment naive, 54 were collected after neoadjuvant or adjuvant therapy but prior to 1L metastatic therapy, and 32 tumors were collected after 1L progression ("post-1L"). The majority (80%) of tumors collected after adjuvant therapy had been exposed to ET, ranging from 6 to 78 months. Duration on adjuvant ET was not significantly associated with tumor ER activity. ER activity progressively decreased through lines of therapy (Fig. 3d), suggesting therapy-induced acquisition of ER-independence. A trend toward longer PFS in the ER high group with giredestrant was consistent across biopsy timings (Supplementary Fig. 7d–f; trend was not apparent with PCET, Supplementary Fig. 7g). Postmenopausal tumors showed significantly lower ER activity than premenopausal tumors (Supplementary Fig. 7h). Half of the post-1L tumors had been previously exposed to CDK4/6i (blue dots, Fig. 3d) but this did not appear to be associated with ER activity.

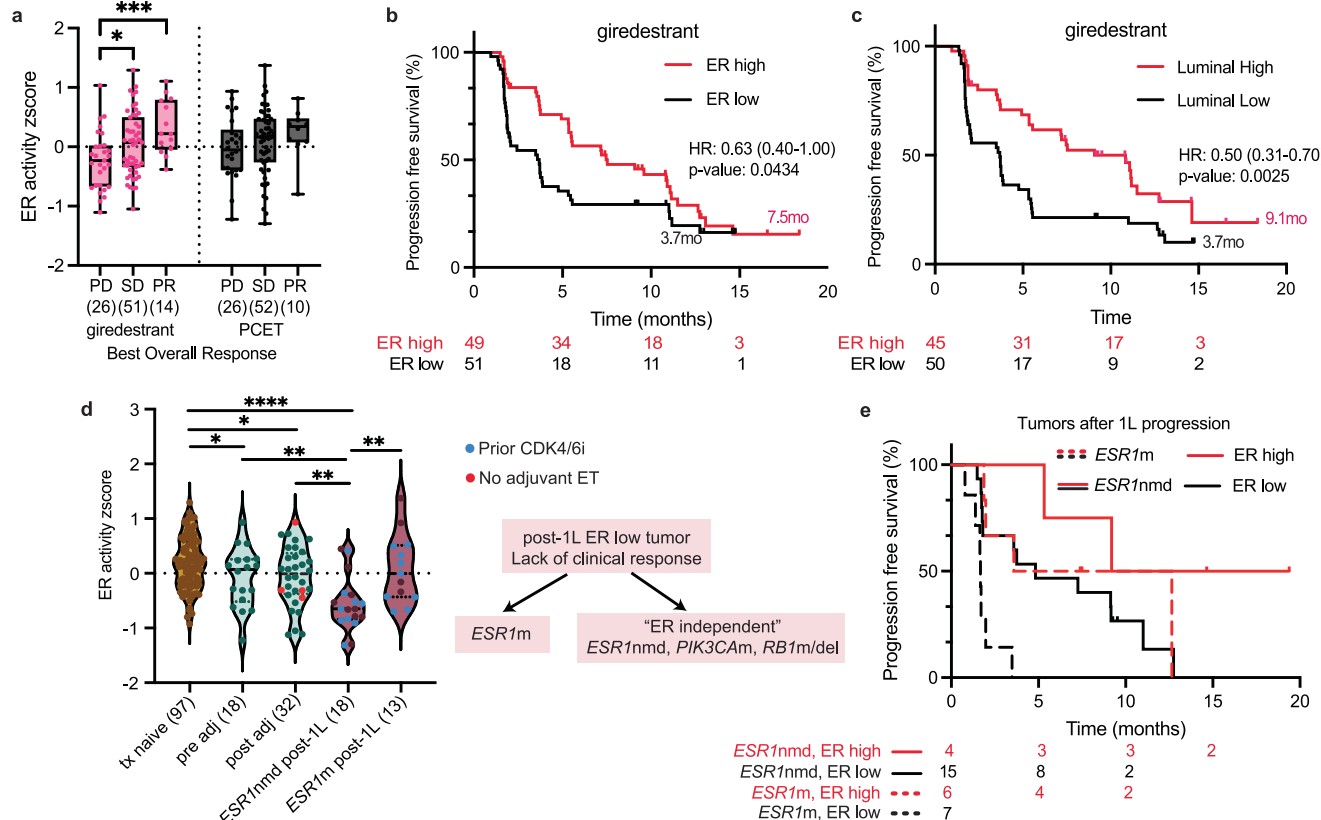

**Fig. 3 | Tumor ER activity is associated with response to giredestrant. a** Baseline tumor ER activity scores (RNA-seq) stratified by treatment arm and best overall response, *$p = 0.0280$, ***$p = 0.0008$. **b** Progression-free survival (PFS) in patients treated with giredestrant, stratified by high (z-score >0) or low (z-score <0) ER activity. **c** PFS stratified by tumor luminal score (above or below median). **d** Baseline tumor ER activity scores split by biopsy collection time relative to prior therapy and *ESR1* mutation status for post-1L samples. Tx naive: tumor has not seen any therapy, pre-adj: tumor was taken after neoadjuvant but before adjuvant therapy, post-adj: tumor was taken after adjuvant therapy, post-1L: tumor was taken after one or two lines of therapy in the advanced setting, *(from left to right) $p = 0.0342, 0.0177$, **(from left to right) $p = 0.0076, 0.0017, 0.003$, ****$p < 0.0001$. **e** PFS for the post-1L cohort, integrated by baseline *ESR1*m status and ER activity z-score. PCET physician's choice endocrine therapy, PD progressive disease, SD stable disease, PR partial response, m mutation, nmd no mutation detected, HR hazard ratio, ER estrogen receptor, adj adjuvant, 1L first line, ET endocrine therapy. Box plots represent median and interquartile range (IQR), with whiskers depicting minimum and maximum. Two-tailed t-tests were used for group comparisons; no multiple-comparison adjustment was performed. PFS p-values were determined by two-sided log-rank (Mantel−Cox) tests. Hazard ratios (HR) and 95% confidence intervals (CI) were calculated using the log-rank approach. Source data are provided as a Source Data file.

After exposure to 1st line therapy, *ESR1*nmd tumors exhibited significantly lower ER activity compared to *ESR1*m tumors (Fig. 3d). *ESR1*m tumors were enriched for myogenesis, hypoxia, and wnt/beta-catenin signaling as well as the Luminal B PAM50 and the integrative cluster (IC) IC3 and IC4 molecular subtypes, consistent with higher ER activity, higher levels of genomic instability, and less favorable outcomes[30–32] (Supplementary Fig. 7i–k). Importantly, a subset of *ESR1*nmd tumors retained high ER activity and achieved clinical benefit (CR, PR, or stable disease for ≥6 months, Fig. 3e), while approximately half of *ESR1*m tumors showed low ER activity and did not benefit from treatment. *PIK3CA* and *RB1* alterations were enriched in "non-responsive" *ESR1*nmd tumors with low ER activity (Supplementary Fig. 7l). These findings suggest that tumor ER activity declines through treatment lines, and that *ESR1*m testing alone is insufficient to identify all patients who may benefit from an oral SERD. The combination of *ESR1*m status and tumor ER activity may provide more comprehensive prediction of treatment response.

## ctDNA dynamics

While baseline genomics can identify prognostic and predictive biomarkers, analysis of longitudinal ctDNA dynamics provides a real-time evaluation of treatment response. To analyze changes in ctDNA during acelERA, we compared baseline and available matched samples at C2D1 (on treatment, $N = 208$ matched) and EOT ($N = 134$ matched).

Overall, 25% (22/89) of patients in the giredestrant arm and 11% (8/75) of patients in the PCET arm switched from ctDNA detected to undetected on treatment. A reduction in ctDNA TF greater than 75% (a cutoff established previously[23]) was associated with improved PFS, and this was significantly more pronounced with giredestrant than PCET (Fig. 4a). ctDNA dynamics appeared to be prognostic independent of the baseline tumor fraction; patients achieving >75% TF reduction had similar baseline TF levels to those who did not, with comparable reduction rates across high and low baseline TF groups. Further, when adjusting for baseline TF in a regression model, a >75% reduction in ctDNA was still significantly predictive of PFS. Patients with PR had lower ctDNA TF levels on-treatment compared to patients with SD or PD (Fig. 4b). ctDNA TF decreased to a greater degree by C2D1 with giredestrant (median −66%) compared to PCET (median −29%) or the fulvestrant subset (median −36%) (Fig. 4c). At EOT, PCET-treated patients, including the fulvestrant subset, showed median TF increases (+20% and +17%, respectively) consistent with progression, while giredestrant-treated patients maintained a median −47% reduction (Fig. 4c). Giredestrant produced greater ctDNA TF declines in *ESR1*m patients (−80%) versus *ESR1*nmd (+5%), a difference not observed with PCET or fulvestrant (Fig. 4d). Similarly, and although small numbers prevent statistical conclusions, post-1L ER high tumors saw a greater median ctDNA TF decline on treatment (−60%) compared to ER low (+3%, Supplementary Fig. 8a). Patients treated with giredestrant

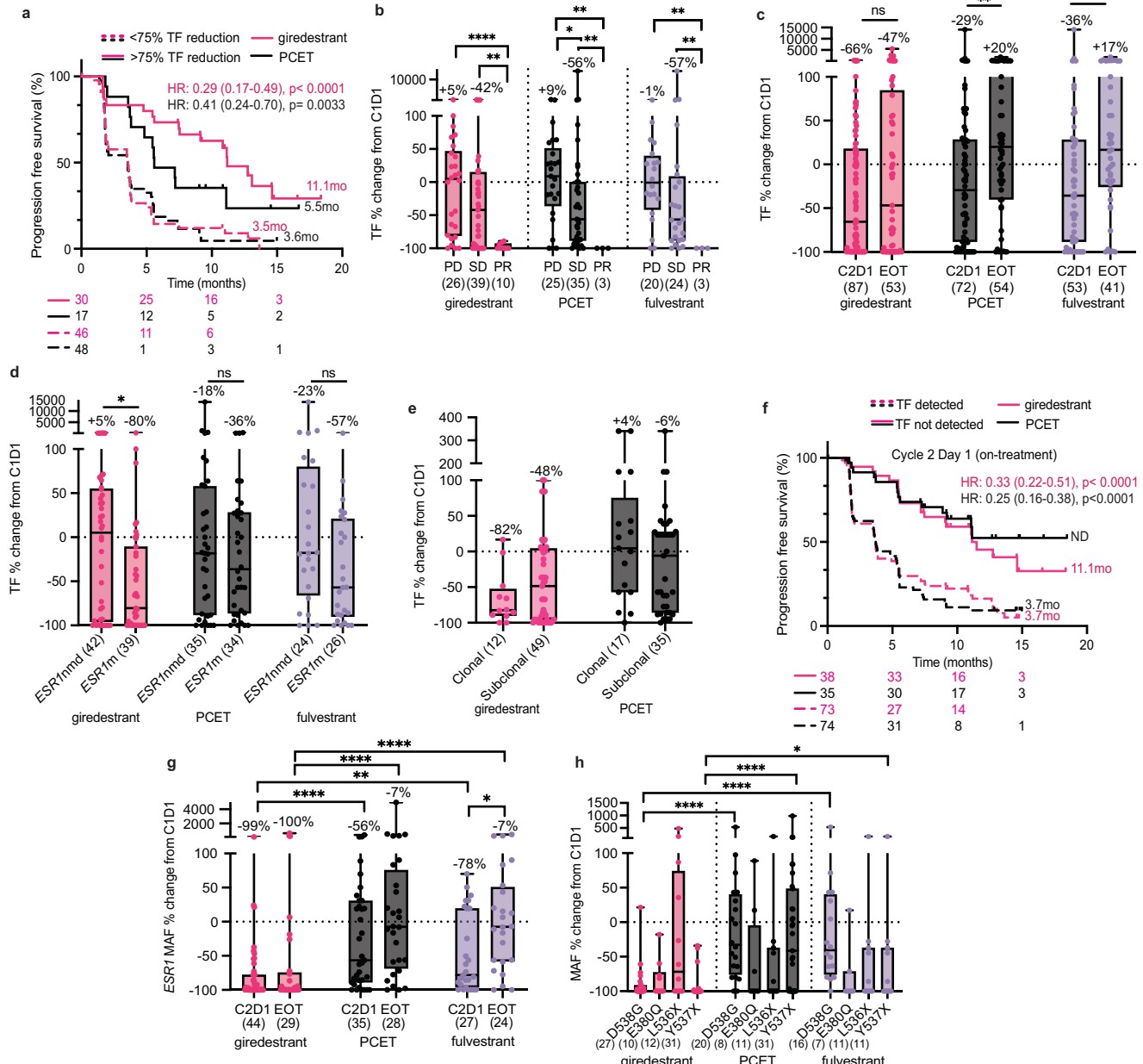

**Fig. 4 | Longitudinal ctDNA dynamics are biomarkers of clinical response and PFS outcomes. a** Progression-free survival (PFS) stratified by 75% reduction in ctDNA tumor fraction (TF) at C2D1. **b** Percent TF change from baseline to C2D1 across treatment arms and confirmed best overall response. Median percentage changes are indicated above each bar; −100% denotes loss of TF detection, *p = 0.0275, ** (from left to right) p = 0.0043, 0.0095, 0.0037, 0.0096, 0.0058, p < 0.0100, ****p < 0.0001. **c** Percent TF change from baseline at C2D1 or end of treatment (EOT) by treatment arm, **(from left to right) p = 0.0052 and 0.054. **d** Percent TF change from baseline at C2D1 stratified by *ESR1* mutation status or **e** *ESR1*m clonality. Samples were defined as 'clonal' if they harbored exclusively clonal *ESR1* variants. 'Subclonal' samples included those with at least one subclonal variant, even in the presence of co-occurring clonal *ESR1* mutations. **f** PFS stratified by ctDNA TF detection status at C2D1. **g** Percent *ESR1* mutant allele frequency

(MAF) change from baseline at C2D1 or EOT by treatment arm. −100% change indicates that *ESR1* MAF went below the limit of detection, *p < 0.0500, **p = 0.0063, ****p < 0.0001. **h** Percent *ESR1* MAF change from baseline at C2D1 by treatment arm and specific *ESR1* variant, *p = 0.0456, ****p < 0.0001. PCET physician's choice endocrine therapy, PD progressive disease, SD stable disease, PR partial response, m mutation, nmd no mutation detected, TF tumor fraction, C2D1 cycle 2 day 1, EOT end of treatment, HR hazard ratio, MAF mutant allele frequency. Box plots represent median and interquartile range (IQR), with whiskers depicting minimum and maximum. Two-sided Mann–Whitney tests were used for group comparisons (p-values indicated). PFS p-values were determined by two-sided log-rank (Mantel–Cox) tests. Hazard ratios (HR) and 95% confidence intervals (CI) were calculated using the log-rank approach. Source data are provided as a Source Data file.

and harboring clonal *ESR1*m tumors also saw greater median levels of TF decline (−82%) compared to those with subclonal *ESR1*m (−48%), suggesting influence of tumor heterogeneity (no statistics due to low sample size, Fig. 4e).

ctDNA detection at C2D1 was prognostic regardless of treatment arm or prior CDK4/6i exposure (Fig. 4f and Supplementary Fig. 8b). ctDNA was undetectable at C2D1 in 66% of patients with clinical benefit

and 75% with confirmed response, versus only 15% without clinical benefit and 28% without confirmed response. These patterns were consistent in both treatment arms. Further, ctDNA detection at C2D1 effectively risk-stratified patients that exhibited stable disease at the 8 week (C3D1) tumor scan, providing valuable granularity to image-based response metrics (HR: 0.30 CI: 0.15–0.63, Supplementary Fig. 8c). The predictive value of ctDNA detection for PFS events changed over time

(Supplementary Fig. 8d), with C2D1 detection showing ~75% positive predictive value for progression within 6 months, while undetectable ctDNA had ~75% negative predictive value. These findings demonstrate the prognostic value of early on-treatment ctDNA assessment, though the 25% progression rate in ctDNA-negative patients suggests continued monitoring is warranted. This highlights the clinical utility of early on-treatment ctDNA testing to evaluate treatment response in advanced breast cancer, and further pinpoint the anti-tumor efficacy of giredestrant in tumors with high ER activity.

### *ESR1* ctDNA dynamics

We next examined *ESR1*m dynamics in ctDNA to evaluate giredestrant's impact on *ESR1* mutant alleles versus PCET. With giredestrant, *ESR1*m prevalence declined from 51% to 40% of patients on-treatment and remained stable at EOT, while PCET-treated patients showed an increase from 47% to 53% at EOT. Six giredestrant-treated patients developed new pathogenic *ESR1* mutations at EOT (D538G, Y537C/N/S, H536D/Y), along with alterations in *TP53*, *RB1*, *PIK3CA*, and *SMAD4*, highlighting potential resistance mechanisms driving disease progression. Several *ESR1* variants of unknown significance were also detected at EOT, primarily glutamate-to-aspartate substitutions in the ligand binding domain. *ESR1* MAF decline on treatment was significantly greater with giredestrant compared to PCET or fulvestrant at both timepoints (Fig. 4g) and patients with response consistently showed near-complete *ESR1* MAF clearance (Supplementary Fig. 8f, g). Based on preclinical and clinical evidence suggesting variant-specific effects on ER antagonist efficacy[33–35], we examined whether individual *ESR1* mutations responded differently to each therapy. It should be noted, as described in Fig. 1, the majority of *ESR1* mutations co-occurred alongside another *ESR1*m. While D538G, E380Q, L536X, and Y537X mutations showed similar patterns of clearance with giredestrant, D538G and Y537X demonstrated significantly greater reduction compared to PCET. (Fig. 4h). Together these results suggest that giredestrant is more effective in clearing tumor cells harboring *ESR1* mutations compared to AIs or fulvestrant, and provide clinical evidence supporting the mechanistic understanding that these constitutively active *ESR1* mutations require an oral SERD approach.

### Clinical, ctDNA, and tumor features associated with rapid clinical progression on endocrine therapy

We observed a sharp initial PFS dropoff in both treatment arms, with 29% (87/303) of patients progressing within 3 months ("rapid progression" or RP), balanced across arms and regardless of *ESR1*m status or prior CDK4/6i exposure (Fig. 5a, b, Supplementary Fig. 9a–c). This "rapid progression" (RP) phenomenon has been noted in other late line ER+ patients receiving ET, including oral SERDs[12–14]. To identify patients unlikely to respond to ET in this setting, we performed an unsupervised analysis of baseline genomic factors which revealed that baseline ctDNA TF and lower tumor ER activity were significantly associated with RP (Supplementary Fig. 9d). Clinical features associated with RP included prior CDK4/6i therapy, duration of prior therapy, liver metastases, and metabolic features such as lower baseline BMI (Supplementary Fig. 9d). While prior CDK4/6i exposure was prognostically unfavorable, 40% of RP patients (*N* = 34) were CDK4/6i-naïve, suggesting this feature alone is not sufficient to predict RP. Similarly, ~40% of *ESR1*m patients exhibited RP, indicating *ESR1*m status alone cannot predict giredestrant response (Fig. 5a, Supplementary Fig. 9a–c). Genomically, RP patients showed higher prevalence of *RB1* and *CDH1* mutations (Fig. 5c), while *ESR1*m patients with RP specifically had higher *PIK3CA* and lower *GATA3* mutation rates (Fig. 5d).

Baseline ctDNA TF was significantly higher in patients with RP compared to non-RP, consistent with our observation that baseline TF is poorly prognostic (Fig. 5e, f). On treatment, TF increased in RP patients (+9%) but decreased substantially in non-RP patients (−69%), reinforcing the prognostic value of TF clearance (Fig. 5g), though *ESR1*

MAF decline was similar between groups (Fig. 5h). These findings indicate higher baseline ctDNA TF and lack of on-treatment ctDNA clearance are associated with rapid progression. Unlike *ESR1*m status, lower ER tumor activity predicted RP specifically in giredestrant-treated patients, but not PCET (Fig. 5i). Transcriptionally, tumors from RP patients showed decreased estrogen response gene expression and increased EMT, with modest elevations in angiogenesis, Wnt/Hedgehog/β-catenin, and adipogenesis signaling (Fig. 5j).

Lastly, using this rich dataset, we developed a multivariate model for predicting rapid clinical progression. A conservative logistic elastic net model (alpha = 0.5 and lambda.1se = 0.169) was used to assess the contribution of 25 clinical and genomic features with rapid progression on *n* = 159 subjects with non-missing data. Binary ER activity score, prior CDK4/6i exposure, presence of liver metastases, and detection of ctDNA by F1LCDx were the variables with non-zero coefficients, which collectively predicted RP with an AUC of 0.79 (Fig. 5k–l).

## Discussion

This exploratory analysis of the acelERA Breast Cancer study examined blood and tissue-based biomarkers in ER+/HER2− La/mBC patients receiving 2L/3L endocrine monotherapy. At baseline, ctDNA was detected in 78% of the evaluated patients, with predominant pathogenic mutations in *ESR1*, *PIK3CA*, *TP53*, and *RB1*. This unique cohort of patients with and without prior CDK4/6i therapy allowed us to understand how CDK4/6i influences biomarker profiles, providing valuable understanding of resistance mechanisms and potentially informing future treatment sequencing strategies post-CDK4/6i. *ESR1* and *RB1* mutations were over twice as prevalent in CDK4/6i-pretreated patients, while giredestrant showed superior efficacy in CDK4/6i-naive *ESR1*m patients. The potential impact of CDK4/6i therapy on ER-related biomarkers merits further study, particularly in the context of primary and secondary endocrine resistance. Together these observations enrich our understanding of the genomic landscape of ER+ La/mBC, which has implications in guiding therapeutic strategies.

Baseline ctDNA positivity predicted early progression, and on-treatment ctDNA clearance was associated with a four-fold improvement in PFS. Giredestrant produced larger declines in ctDNA TF and *ESR1*m allele burden than PCET/fulvestrant, mirroring its superior activity in *ESR1*m-mutant tumors. Early ctDNA dynamics, particularly clearance and C2D1 detection, appeared to correlate with PFS and response, underscoring baseline and early on-therapy sampling as actionable decision points. Concurrently, baseline tumor RNAseq revealed ER signaling predicted giredestrant response. Importantly, some high-ER-activity patients responded despite lacking *ESR1*m, while certain *ESR1*m patients with low ER activity responded poorly. This indicates *ESR1*m status alone is insufficient for patient selection and suggests potential benefit from earlier giredestrant intervention when ER activity remains high such as in the 1st line or adjuvant settings. This was evidenced in the lidERA study, in which giredestrant demonstrated benefit in early breast cancer (where *ESR1* mutations are rare)[36].

While our study identifies several promising biomarkers (*ESR1* mutations, tumor ER activity, and ctDNA dynamics), translating these into a practical clinical test presents challenges. Most patients in this study's patient population harbor multiple *ESR1* variants simultaneously, complicating variant-specific biomarker development. Further, tumor biopsies are often not feasible for patients with advanced disease, making direct measurement of tumor ER transcriptional signatures challenging and necessitating blood-based alternatives. We propose development of an epigenomic liquid biopsy-based ER activity classifier that can be integrated with *ESR1* mutation status and ctDNA metrics to comprehensively predict SERD benefit. Emerging technologies such as cell-free DNA chromatin immunoprecipitation (ChIP)[37,38], fragmentomics[39–42], and DNA methylation[43,44] can effectively measure transcriptional programs from blood samples and could potentially provide non-invasive surrogates for tumor ER activity

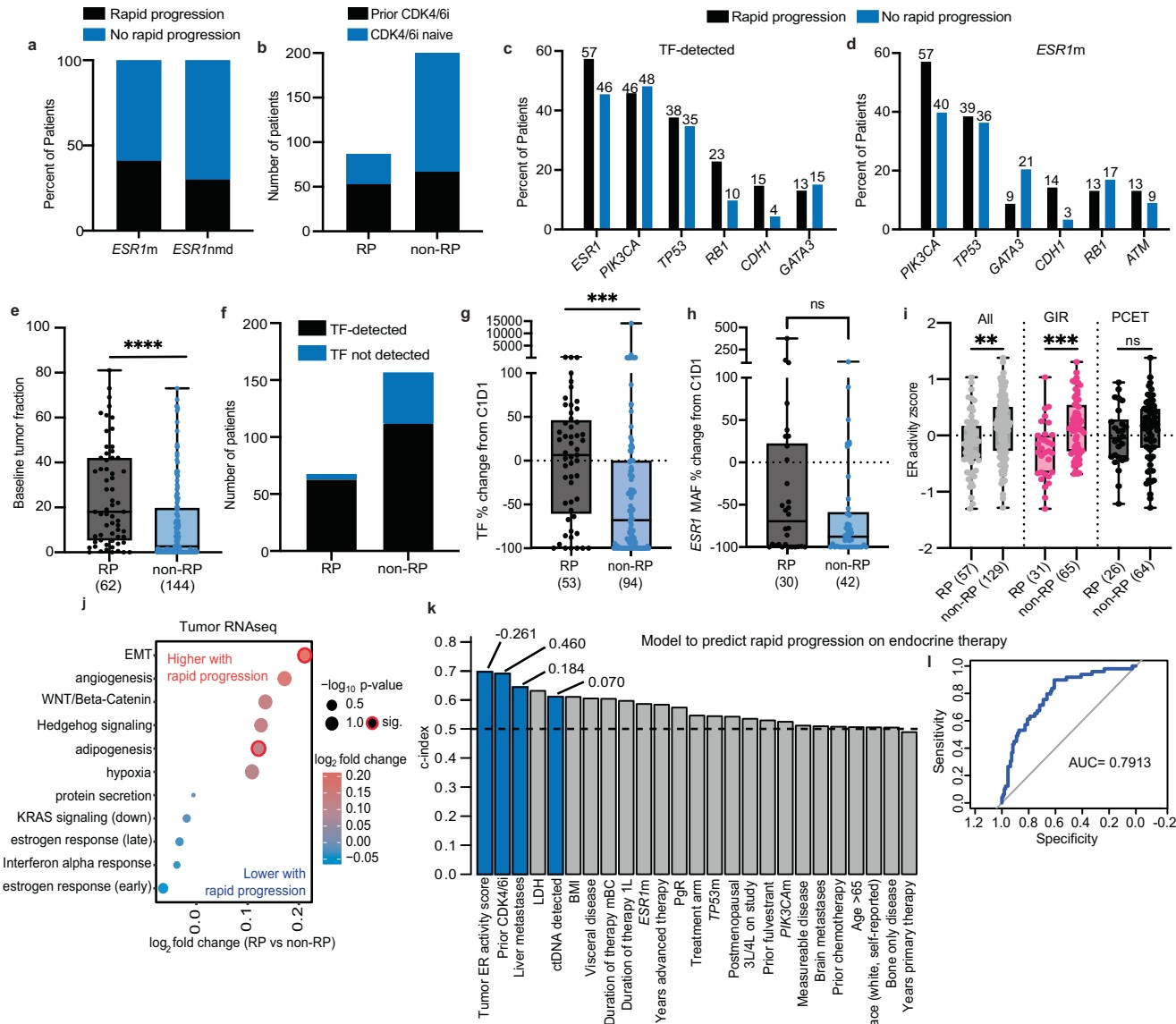

**Fig. 5 | Rapid progression is associated with ctDNA TF, clinical features, and low tumor ER activity. a** Percent of patients that experienced RP, stratified by *ESR1*m status or **b** prior CDK4/6i exposure. **c** Prevalence of common pathogenic short variants by RP status in patients with detectable ctDNA TF or **d** patients in the *ESR1*m subgroup. **e** Baseline ctDNA tumor fraction (TF, ****$p < 0.0001$) and **f** ctDNA detection rates by RP status. **g** Percent change from baseline at C2D1 for ctDNA TF (***$p = 0.0005$) and **h** *ESR1* mutant allele frequency (MAF) by RP status, $p = 0.1378$. **i** Tumor ER activity score by treatment arm and RP status, $p = 0.3211$, **$p = 0.0011$, ***$p = 0.0003$. **j** Differential enrichment of Hallmark signatures (GSVA) associated with RP. The top six pathways in each direction are shown; significance was determined using moderated linear models (limma), unadjusted for multiple

comparisons. **k** Concordance index (c-index) for variables evaluated for RP prediction. Blue bars denote features selected by a logistic elastic net model (beta values indicated). **l** Receiver-Operating Characteristic (ROC) curve of the final model for predicting RP. GIR giredestrant, PCET physician's choice endocrine therapy, m mutation, nmd no mutation detected, RP rapid progression, TF tumor fraction, ER estrogen receptor, MAF mutant allele frequency, AUC area under curve, GSVA gene set variation analysis. Box plots represent median and interquartile range (IQR), with whiskers depicting minimum and maximum. Two-tailed $t$-tests were used for ER activity comparisons (i); two-sided Mann−Whitney tests were used for ctDNA-related comparisons (**e**, **g**, **h**); no multiple-comparison adjustment was performed. Source data are provided as a Source Data file.

assessment, enabling more precise patient selection beyond *ESR1*m status alone.

Importantly, our findings have significant implications for future clinical trial design. Our analysis showed decreased ER activity in post-1L *ESR1*nmd tumors, potentially explaining variable SERD response in these patients. This observation, combined with identified actionable targets like *PIK3CA*m, supports investigating SERD combinations with PI3K/AKT inhibitors. The identification of ER activity as a key predictor of SERD response provides a potential stratification factor for upcoming studies, particularly those evaluating oral SERDs in combination with next-generation CDK inhibitors or other targeted agents. Our ctDNA dynamics data also offer a framework for early response

assessment that could facilitate more efficient development of these combinations. A multivariable model combining ER activity score, ctDNA status, liver metastasis, and prior CDK4/6i exposure identified rapid progressors (AUC = 0.79), providing a potentially powerful tool for identifying patients unlikely to respond to ET. Together, these findings show that ctDNA and tumor transcriptomics provide complementary prognostic and predictive information for optimizing SERD therapy selection and adaptation.

Limitations of this study include unpowered subgroups that were not predefined, potential clonal hematopoiesis[45] confounding *TP53* findings, limited paired tumor/ctDNA samples, tumor heterogeneity, and varied prior therapies. Without multiple comparison adjustment,

findings remain hypothesis-generating. Cross-trial analyses with real-world data are needed for confirmation. Nevertheless, these findings provide insight into the mutation landscape of late line ER+ La/mBC patients and highlight the utility of blood and tumor-based biomarkers to predict and evaluate therapy response to guide future development of oral SERD precision medicine strategies.

## Methods

### Ethics & inclusion statement

This retrospective study adhered to the REMARK guidelines. The study did not result in any stigmatization, incrimination, or discrimination to participants. acelERA BC was performed in accordance with Good Clinical Practice guidelines and the Declaration of Helsinki. Protocol approval was obtained from an independent ethics committee for each participating site. Every patient gave written informed consent.

### Patient population

acelERA BC study design, patients, oversight, and assessments were described previously[15]. Briefly, the patient population was post- or pre-/perimenopausal women or men, ≥18 years with measurable or bone-only disease who progressed after 1–2 lines of systemic therapy. Patients were randomized 1:1 to giredestrant or fulvestrant/AI and treated until disease progression or unacceptable toxicity. Patients were stratified by visceral disease, prior CDK4/6i, and prior fulvestrant. Prior treatment with an investigational SERD was not allowed. The clinical cutoff date used for these analyses was July 15, 2022. Clinical benefit rate (CBR) was defined as a confirmed complete or partial response or stable disease for ≥6 months. Overall response rate (ORR) was defined as a confirmed complete or partial response. As reported previously, 302/303 participants in acelERA were female, so biomarker data from a single male participant are included throughout this manuscript. Due to the insufficient sample size, sex-based analyses were not reported.

### Evaluation of plasma samples

F1LCDx was performed in the Foundation Medicine Inc. laboratory, according to standard workflow[46]. Plasma was separated from whole blood by centrifugation and cfDNA was isolated with a KingFisher™ Flex Magnetic Particle Processor (Thermo Fisher, Waltham, MA, USA) and constructed into genomic libraries. Libraries were sequenced on the Illumina NovaSeq 6000 platform (Illumina Inc., San Diego, CA, USA). Samples with sufficient cfDNA content (lowest cfDNA input mass yielding successful results was 24.34 ng), as assessed by the Agilent 4200 TapeStation assay, were used for further analyses. For longitudinal analysis of changes in ctDNA levels, results were reanalyzed for computational bridging to FoundationOne®Monitor, an assay intended to report ctDNA tumor fraction quantity longitudinally for monitoring response to therapy.

Pathogenic short variants in this study were defined as short nucleotide variants with known or likely impact on protein function (per COSMIC). ctDNA tumor fraction (TF) quantification provides an estimate of the percentage of ctDNA present in a cfDNA sample[47]. The TF algorithm used a multi-omic assessment from cell free DNA and filtered suspected clonal hematopoiesis-derived alterations algorithmically. All samples with ctDNA TF detected (TF > 0) were considered "TF-detected". The lowest ctDNA TF value detected and quantified across all samples was 0.15%. Patients with ctDNA detected but not quantifiable, as well as patients without detectable ctDNA at both time points were excluded from percent change analyses. Variant clonality was calculated as MAF divided by TF, and variants with MAF/TF < 0.25 were considered sub-clonal. These analyses were performed by Foundation Medicine as a part of the F1LCDx data output. ESR1 mutation status for patients enrolled in China was assessed using the PredicineCARE liquid biopsy assay. ESR1 mutations were classified as 'clonal' only if all detected ESR1 mutations

were clonal. Conversely, samples were categorized as 'subclonal' if at least one subclonal ESR1 variant was present, regardless of whether additional clonal variants of the same or different ESR1 mutations were identified.

The PPV and NPV of ctDNA detection in predicting PFS events were estimated at multiple intervals after the C2D1 F1LCDx test using the cumulative incidence function estimator. The time to a PFS event after F1LCDx was defined as the number of months from C2D1 to a PFS event, censored at the last follow-up. Clustered bootstrapping was used to account for the nonindependence of multiple tests per patient.

Baseline TF and the categorical variable of whether a patient experienced 75% clearance were tested for their relationship with PFS by fitting a cox proportional hazards regression model with either each as a single variable or with both together (PFS ~baseline TF + 75% clearance) in the subset of patients with both data types. This was performed in R using coxph() [survival v3.8-3].

### Tumor RNAseq

Tumor RNAseq was performed by Q2 Lab Solutions using the RNA Access Library Preparation Kit (Illumina) followed by paired-end (2 × 50b, 50 M reads) sequencing on the HiSeq platform (Illumina). RNASeq reads were aligned to the human reference genome (NCBI Build 38) using GSNAP (Wu and Nacu; Wu et al.) version 2013-10-10, allowing a maximum of two mismatches per 75 base sequence (parameters: '-M 2 -n 10 -B 2 -i 1 -N 1 -w 200000 -E 1-pairmax-rna = 200000 –clip-overlap). To quantify gene expression levels, the number of reads mapped to the exons of each RefSeq gene was calculated using the functionality provided by the R/Bioconductor package GenomicAlignments. Raw counts were normalized to transcripts-per-million (TPM). ER z-scores were calculated using a predefined, experimentally derived set of 38 ER target genes[18] normalized to a reference set of ER+ breast cancer samples. Voom was used to identify differentially expressed genes and generate count tables from raw reads which were used to derive PAM50 and IC10[32] subtypes. For gene pathway analysis, hallmark gene sets were downloaded from MSigDB (http://www.gsea-msigdb.org/gsea/msigdb/human/collections.jsp#H; v2022.1.Hs) and enrichment scores were determined using gsva (R package: GSVA [v1.50.5]; options min.sz=2, max.sz=1000). For analyses of hallmark signatures with PFS, each signature was evaluated for association with PFS with a Cox proportional-hazards model (Cox et al.) using the coxph function (R package: survival [v3.8-3]), and significance was assessed also with this function using a Wald test. For association of hallmark signatures with categorical variables such as ESR1 mutation status or rapid progression, we fit a univariate linear model using lmFit and then applied empirical Bayes moderation (R package limma [v3.58.1]). Heatmaps for visualization were generated using the Heatmap function (R package ComplexHeatmap [v2.18.0]). Dot plots created with ggplot (R package ggplot2 [v3.5.1]) and data manipulation assisted by R package dplyr [v1.1.4].

### Statistics and reproducibility

ctDNA data (other than ESR1 mutation status) from patients with PredcineCARE results were excluded from most analyses in this manuscript as this assay did not provide tumor fraction calculation and is not directly comparable to the F1LCDx results. No other data were excluded from these analyses. Replication was not performed as this is not applicable for clinical data. The statistical methods used to determine study size, as well as randomization and blinding procedures associated with the clinical trial are described in the acelERA primary clinical manuscript. Technical replicates were not performed for ctDNA or tumor RNAseq due to limited sample material. Almost all patients evaluated in this study were female (since this study focused on hormone-receptor positive breast cancer, a disease that primarily affects female patients). These demographics were reported in the primary clinical manuscript. Patient sex was determined by

an evaluating physician as part of the clinical trial. Patient gender was not evaluated or collected. Disaggregated sex data is not applicable. Group differences between percent changes or ctDNA levels were calculated using the non-parametric two-sided Mann–Whitney test. Cohorts with normal distributions were evaluated using a two-tailed (two-sided) $t$-test. Progression-free survival (PFS) was estimated using the Kaplan-Meier method and statistical significance was determined by the two-sided log-rank (Mantel-Cox) test. Hazard ratios (HR) and 95% confidence intervals (CI) were calculated using the log-rank approach. Group comparison and PFS statistical analyses were performed in GraphPad Prism (v10.1.1). Hazard ratios and confidence intervals for exploratory subgroup analysis were estimated using Cox proportional-hazards models using the coxph function (R package survival [v3.8-3]). Visualization as a forest plot was done using the forestplot function (R package forestplot [v3.1.6]) or the forest_model function (R package forestmodel [v0.6.2]). All measurements were taken from distinct samples/patients and were not measured repeatedly. Due to limited sample size and the exploratory nature of this study, and the fact that only a single pre-specified comparison was conducted in these instances, adjustment for multiple hypothesis testing was not performed unless noted otherwise.

### Rapid progression prediction model

cv.glmnet() from the glmnet (v4.1-8) R package with family = "binary" and alpha = 0.5 were used for the logistic elastic net modeling. pROC was used for AUC. Bone only disease and duration of 2L treatment were dropped due to high levels of missing data. PAM50 and ancestry were dropped due to low N in subgroups influencing results after first confirming the major subgroups had low predictive value. After dropping patients missing data for the remaining clinical features, $n = 159$ patients remained, with 25 variables.

### Reporting summary

Further information on research design is available in the Nature Portfolio Reporting Summary linked to this article.

## Data availability

The Source Data file contains derived, non-identifying data but does not include individual patient-level clinical records. Anonymized, publication-specific biomarker datasets, minimized to those necessary to reproduce the analyses reported here, may be made available upon request and subject to review and approval by Roche, including execution of a data sharing agreement. For up-to-date details on Roche's Global Policy on the Sharing of Clinical Information and how to request access to related clinical study documents, see https://go.roche.com/data_sharing. The study protocol is available with the previously published primary analysis (DOI:10.1200/JCO.23.01500). For eligible studies, qualified researchers may request access to individual patient-level clinical trial data underlying the primary study endpoints through a data request platform. At the time of writing, this request platform is Vivli: https://vivli.org/ourmember/roche/. For up-to-date details on Roche's Global Policy on the Sharing of Clinical Information and how to request access to related clinical study documents, see here: https://go.roche.com/data_sharing. Anonymized records for individual patients across more than one data source external to Roche cannot, and should not, be linked due to a potential increase in risk of patient reidentification. Source data are provided with this paper.

## Code availability

Custom code or mathematical algorithms were not developed in this study. All analyses were completed using publicly available R packages (see Methods and Reporting Summary for versions) or using Prism (v10.1.1, see Statistical Methods).

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

## Acknowledgements

We would like to thank Steven Gendreau, Marc Hafner, Jackson Liang, Ciara Metcalfe, and Matthew Wongchenko for their review and discussion. This research was funded by F. Hoffmann-La Roche Ltd. F. Hoffmann-La Roche Ltd was involved in the study design, data interpretation, and decision to submit for publication in conjunction with the authors.

## Author contributions

A.E.C. wrote the manuscript. A.E.C and S.H. constructed the figures. A.E.C., S.H., L.W.P., T.M.F., and H.M.M. designed, performed, and interpreted the analyses. A.M.C. and X.S. provided additional computational support and data integration. M.CM., A.B., M.M., E.L, J.S., and P.D.PM. conducted the clinical study. C.D., P.D.PM., and H.M.M. conceived and executed the biomarker sampling plan for the clinical study. All authors reviewed the manuscript.

## Competing interests

The study was sponsored by F. Hoffmann-La Roche Ltd. The authors declare the following competing interests: A.E.C., S.H., A.M.C., P.D.P.-M., and H.M.M. are employees of Genentech, Inc. and hold stock in F. Hoffmann-La Roche Ltd. C.D. is an employee of Roche (China) Holding Ltd. and holds stock in F. Hoffmann-La Roche Ltd. and Bristol Myers Squibb. L.W.P. is an employee of Foundation Medicine and holds stock in Roche Holdings AG. M.CM. reports consulting fees from Novartis, AstraZeneca, and Lilly; honoraria from Exact Sciences; travel support from Novartis and AstraZeneca; advisory board participation for Astra-Zeneca and Roche/Genentech; and board membership for Legacy Community Health, The Hope Foundation, and ASCO. A.B. reports grants and personal fees from Pfizer, Novartis, Genentech, Merck, Menarini, Gilead, Sanofi, Daiichi Pharma/AstraZeneca, Alyssum, and Eli Lilly. M.M. reports honoraria and advisory fees from Roche/Genentech, Lilly, Pfizer, Novartis, Pierre Fabre, Seagen, AstraZeneca, and Daiichi-Sankyo; and research funding from Novartis, Roche, and Puma Biotechnology. E.L. reports institutional advisory/steering committee roles and research funding from AstraZeneca, Gilead, Lilly, MSD, Novartis, Pfizer, and Roche; royalties from Walter and Eliza Hall Institute; and leadership roles at Breast Cancer Trials Australia, Garvan Institute, St Vincent's Hospital, and UNSW. J.S. reports institutional research funding from Seagen, MSD, Roche, Pfizer, Novartis, AstraZeneca, Lilly, GSK,

Boehringer Ingelheim, Sanofi, Daiichi Sankyo, Qurient, Dragonfly, Eikon, Gilead, Celcuity, BMS, HLB Life Science, Sermonix, Olema, Hanmi, Ildong, and Samyang; and family stock ownership in Daiichi Sankyo. X.S. declares no competing interests.
