## [Transparent Peer Review file · Nature Communications]

ctDNA and tumor-based biomarkers of giredestrant response in aceLERA Breast Cancer

Corresponding Author: Dr Ann Collier

Version 0:

Reviewer comments:

Reviewer #1

(Remarks to the Author)

I reviewed this manuscript during a previous submission for a different journal. The authors did a good job of addressing my concern and I would recommend publication. The only remaining issue is that Supplemental Fig 5a seems to have incorrect numbers under "genetic ancestry". I suspect the columns are swapped for "TF detected" and "TF not detected", because the sum of the "TF detected" category matches the total number of patient with TF *NOT* detected (N=50). I'd ask that the authors please double check all numbers in tables to make sure they are correct.

Reviewer #2

(Remarks to the Author)

The authors have adequately addressed my comments.

Reviewer #3

(Remarks to the Author)

It is nice to see that the authors have made significant effort to perform thoughtful and rigorous revisions the authors have made to address key points raised in our initial review, but we have remaining concerns that should be addressed given the short turnaround time of their response.

This manuscript presents a compelling and timely correlative science study in the context of an already reported clinical trial. While the study proposes clinically actionable biomarkers, most notably ER transcriptional activity and ctDNA-based response metrics (C2D1), that have the potential to improve patient outcomes using tools already accessible in many clinical settings, the conclusions may need to be moderated until others are able to replicate their finding or reanalyze the data when publicly available. Nonetheless, the expanded discussion on mechanistic drivers of endocrine sensitivity, real-time ctDNA monitoring, and the feasibility of surrogate assays for ER activity further enhances the translational relevance of this work. This study adds important evidence specific to HR+ breast cancer, demonstrating that early ctDNA clearance and ESR1 MAF reduction (C2D1) can serve as meaningful predictors of response. This complements findings from other cancers (ie. TRACERx, DYNAMIC, and ctMoniTR), where early ctDNA dynamics vary more widely (30–70%) depending on tumor type and treatment modality. The heterogeneity of breast cancer in diverse populations makes this a very important proof of concept paper given the differential overall survival of Black women with HR+ breast cancer but the numbers are small to come to definitive conclusions about predictive value of their finding even though this was a global study. The authors could take time to address the following or discuss why it is not within scope of current study.

1. Tumor ER activity is proposed as a biomarker specific to giredestrant response. That is an overstatement because the survival curves shown in Figure S7b show similar association of ER activity / ER high/low disease with response to giredestrant / PCET. Even though the survival curves in Figure S7b did not reach significant difference, the hazard ratio is actually lower in the PCET subgroups with $p = 0.05$. A more appropriate way to test interaction of giredestrant treatment with ER high / low would be to fit a single Cox model with treatment received, ER high/low status, and an interaction term - if the

interaction term is significant, one could conclude that ER high is specifically a biomarker of response to giredestrant (as stated in the manuscript)- but the data presented do not suggest that ER high is predictive of ONLY giredestrant benefit. Similarly, for ER activity z-score (Figure 5a), a trend towards higher z-score with better response is seen in both treatment groups. A two-way ANOVA or equivalent model should be used to check interaction of response category with treatment received before performing pairwise t-tests between groups - and you should only state that ER activity z score differences are specific to giredestrant if the treatment x response interaction F test is significant.

2. Along those lines, although a formal predictive model for the Rapidly Progressing (RP) subgroup was not developed, the authors provided meaningful exploratory insights and articulated future directions. This is the most clinically significant subgroup! Given the richness of the dataset, a multivariable risk model remains feasible and could offer substantial value in guiding early treatment strategies, especially for Luminal B tumors before they become hormone refractory.

3. Supplemental figure 4 - median PFS values in subgroups would be beneficial to list.

4. For a Pharma led studies, it would be useful if summary data could be deposited into the public domain for secondary analysis if possible.

Version 1:

Reviewer comments:

Reviewer #4

(Remarks to the Author)

Reviewer #5

(Remarks to the Author)

Reviewer comments are copied below in black

Responses to Reviewers are highlighted in blue

Edits in the manuscript are highlighted in red (revision #1) or pink (revision #2)

Reviewer #1 (Remarks to the Author):

I reviewed this manuscript during a previous submission for a different journal. The authors did a good job of addressing my concern and I would recommend publication. The only remaining issue is that Supplemental Fig 5a seems to have incorrect numbers under "genetic ancestry". I suspect the columns are swapped for "TF detected" and "TF not detected", because the sum of the "TF detected" category matches the total number of patient with TF *NOT* detected (N=50). I'd ask that the authors please double check all numbers in tables to make sure they are correct.

Thank you for your thoughtful review of our work and for catching this error. We have edited the table and double-checked the other results.

Reviewer #2 (Remarks to the Author):

The authors have adequately addressed my comments.

Thank you for taking the time to review our manuscript.

Reviewer #3 (Remarks to the Author):

It is nice to see that the authors have made significant effort to perform thoughtful and rigorous revisions the authors have made to address key points raised in our initial review, but we have remaining concerns that should be addressed given the short turnaround time of their response.

Thank you for your thoughtful review of the work. The reviews from Nature Cancer were received on May 6th, and a revision was sent to Nature Communications on July 9th. Revisions took approximately two months, but no additional lab-based experiments were performed.

This manuscript presents a compelling and timely correlative science study in the context of an already reported clinical trial. While the study proposes clinically actionable biomarkers, most notably ER transcriptional activity and ctDNA-based response metrics(C2D1), that have the potential to improve patient outcomes using tools already accessible in many clinical settings, the conclusions may need to be moderated until others are able to replicate their finding or reanalyze the data when publicly available. Nonetheless, the expanded discussion on mechanistic drivers of endocrine sensitivity, real-time ctDNA monitoring, and the feasibility of surrogate assays for ER activity further enhances the translational relevance of this work. This study adds important evidence specific to HR+ breast cancer, demonstrating that early ctDNA clearance and ESR1 MAF reduction (C2D1) can serve as meaningful predictors of response.

This complements findings from other cancers (ie. TRACERx, DYNAMIC, and ctMoniTR), where early ctDNA dynamics vary more widely (30–70%) depending on tumor type and treatment modality. The heterogeneity of breast cancer in diverse populations makes this a very important proof of concept paper given the differential overall survival of Black women with HR+ breast cancer but the numbers are small to come to definitive conclusions about predictive value of their finding even though this was a global study. The authors could take time to address the following or discuss why it is not within scope of current study.

We appreciate the reviewer's thoughtful summary and fully agree that our data, while encouraging, remain hypothesis-generating until validated externally (ongoing in larger ph3 studies at Roche and elsewhere). We have therefore tempered the language of our key conclusions, added explicit caveats, and flagged future validation steps (see manuscript text highlighted throughout in pink).

There is a complementary manuscript from our group currently under review at Nature Communications (Liang et al. "*Loss of luminal lineage drives resistance to next-generation ER α -antagonists in pretreated ER + HER2 - locally-advanced or metastatic breast cancer*"). Similar to what we see in acELERA, this study demonstrates that giredestrant-responsive tumors maintain high dependency on ER signaling and that non-responsive tumors exhibit loss of luminal lineage identity and ER dependence. This manuscript includes follow-up *in vitro* validation with giredestrant-resistant cell lines and multi-omic NGS that provide further insight into ER-independence mechanisms. **We have suggested to the editorial staff that these manuscripts be co-published, if accepted.**

1. Tumor ER activity is proposed as a biomarker specific to giredestrant response. That is an overstatement because the survival curves shown in Figure S7b show similar association of ER activity / ER high/low disease with response to giredestrant / PCET. Even though the survival curves in Figure S7b did not reach significant difference, the hazard ratio is actually lower in the PCET subgroups with $p = 0.05$. A more appropriate way to test interaction of giredestrant treatment with ER high / low would be to **fit a single Cox model with treatment received, ER high/low status, and an interaction term** - if the interaction term is significant, one could conclude that ER high is specifically a biomarker of response to giredestrant (as stated in the manuscript)- but the data presented do not suggest that ER high is predictive of ONLY giredestrant benefit. Similarly, for ER activity z-score (Figure 5a), a trend towards higher z-score with better response is seen in both treatment groups. **A two-way ANOVA or equivalent model should be used to check interaction of response category with treatment received before performing pairwise t-tests between groups** - and you should only state that ER activity z score differences are specific to giredestrant if the treatment x response interaction F test is significant.

Thank you for noting the statistical limitations in the previous revision. The Reviewer is correct that PCET and giredestrant hazard ratios for ER high vs. low are similar (0.61 and 0.63). Median PFS with PCET was similar between groups (5.55 vs. 5.14 months), whereas giredestrant caused longer PFS in ER high (7.52 vs. 3.68 months). However, hazard ratios suggest ER high patients may benefit regardless of treatment. **We have removed "specifically" in the relevant manuscript sections and noted the trends for PCET.**

The reviewer is correct that both treatment groups show similar directionality (higher ER in responders), but the PCET cohort did not reach statistical significance. The Reviewer's suggested Cox model and two-way ANOVA revealed that the treatment x response interaction was not significant ($p=0.344$ and 0.946 , respectively). It should be noted that these studies were not powered to test for interaction, and a lack of significance should be interpreted with caution. It should also be noted that the key conclusion still remains – patients with high tumor ER activity receive increased clinical benefit from giredestrant.

We have edited the text as such: *“ER transcriptional activity...was significantly higher in tumors that exhibited stable disease (SD) or PR with giredestrant; PCET showed a similar direction of effect but did not reach significance. A non-significant ER activity × treatment arm interaction ($p = 0.946$ in Cox model and $p=0.344$ in two-way ANOVA) suggests ER activity's predictive effect may apply across ETs, though this study was not powered for interaction analysis, an effect not observed with PCET. Patients with high ER tumor activity (z-score >0) or more luminal-like tumors...also showed longer PFS specifically with giredestrant (median 7.5 vs 3.6 months. The PCET arm showed a similar but non-significant separation (5.6 vs 5.1 months).”*

As ER activity can diminish with successive therapy lines (Fig. 3d), use of archival biopsies means some tumors labeled “ER-high” at enrollment could have been ER-low at treatment start, potentially diluting the true PFS advantage for ER-high disease. This complicates comparison between giredestrant and PCET. Restricting to freshly collected tumors (**newly added Fig. S7f-g**) yields a better hazard ratio for giredestrant (0.56 vs. 0.68), but small numbers and overlapping confidence intervals warrant caution. **We added the following to the results section:** *“A key caveat is that many tumors were scored from archival biopsies; if ER activity declined during intervening treatments, some “ER-high” cases were probably mis-classified as such, dampening the true benefit signal and confounding any direct comparison between giredestrant and PCET.”*

2. Along those lines, although a formal predictive model for the Rapidly Progressing (RP) subgroup was not developed, the authors provided meaningful exploratory insights and articulated future directions. This is the most clinically significant subgroup! Given the richness of the dataset, a multivariable risk model remains feasible and could offer substantial value in guiding early treatment strategies, especially for Luminal B tumors before they become hormone refractory.

We appreciate the Reviewer's suggestion and agree this is a worthwhile addition to the manuscript. We developed a formal prediction model for rapid progression using logistic elastic net. Out of 25 variables considered, the final model picked tumor ER activity, prior CDK4/6i exposure, presence of visceral mets, and ctDNA detection by F1LCDx. Together the model predicted RP with an AUC of 0.79. **c-index for each feature considered as well as the model ROC curve have been added to Figure 5.**

3. Supplemental figure 4 - median PFS values in subgroups would be beneficial to list.

We have now added this information to both Supplemental Figures 3 and 4.

4. For a Pharma led studies, it would be useful if summary data could be deposited into the public domain for secondary analysis if possible.

The source data for all figures have been provided as a Source Data file. This file includes per-patient tumor ER activity, PAM50, IC10, and luminal scores, all baseline alterations, longitudinal tumor fraction, key baseline demographics, BOR, etc. Any other data supporting the findings of this study may be available from the corresponding author on reasonable request. For eligible studies, qualified researchers may request access to individual patient-level clinical data through a data request platform. For up-to-date details on Roche's Global Policy on the Sharing of Clinical Information and how to request access to related clinical study documents, see here: https://go.roche.com/data_sharing. Anonymized records for individual patients across more than one data source external to Roche cannot, and should not, be linked due to a potential increase in risk of patient reidentification.